# Renal graft function in transplanted patients correlates with CD45RC T cell phenotypic signature

**Séverine Bézie[1]\*, Céline Sérazin[1], Elodie Autrusseau[1], Nadège Vimond[1], Magali Giral[1,2], Ignacio Anegon[1], Carole Guillonneau[1]\***

**1** Center for Research in Transplantation and Translational Immunology, Nantes Université, INSERM, UMR 1064, F-44000, Nantes, France, **2** Department of Nephrology, CHU Nantes, Nantes Université, ITUN, Nantes, France

\* carole.guillonneau@univ-nantes.fr (CG); severine.bezie@univ-nantes.fr (SB)

**Data Availability Statement:** All relevant data are within the manuscript and its Supporting information files.

## Abstract

Biomarkers that could predict the evolution of the graft in transplanted patients and that could allow to adapt the care of the patients would be an invaluable tool. Additionally, certain biomarkers can be target of treatments and help to stratify patients. Potential effective biomarkers have been identified but still need to be confirmed. CD45RC, one of the splicing variants of the CD45 molecule, a tyrosine phosphatase that is critical in negatively or positively regulating the TCR and the BCR signaling, is one marker already described. The frequency of CD8+ T cells expressing high levels of CD45RC before transplantation is increased in patients with an increased risk of acute rejection. However, single biomarkers have limited predictive reliability and the correlation of the expression levels of CD45RC with other cell markers was not reported. In this study, we performed a fluorescent-based high dimensional immunophenotyping of T cells on a cohort of 69 kidney transplant patients either with stable graft function or having experienced acute transplant rejection during the first year after transplantation or at the time of rejection. We identified combinations of markers and cell subsets associated with activation/inflammation or Tregs/tolerance (HLA-DR, PD-1, IFNγ, CD28) as significant biomarkers associated to transplant outcome, and showed the importance of cell segregation based on the CD45RC marker to identify the signature of a stable graft function. Our study highlights potential reliable biomarkers in transplantation to predict and/or monitor easily graft-directed immune responses and adapt immunosuppression treatments to mitigate adverse effects.

## Introduction

Significant progresses have been made in the past decades to efficiently control acute kidney allograft rejection thanks to the long-term use of non-specific immuno-suppressant drugs (IS). However these IS induce numerous adverse events, such as off effects on blood pressure or cholesterol metabolism and others due to wide-spread immunosuppression, such as cancer and infections, affecting quality of life and life expectancy of the patients [1]. Furthermore, IS

**Funding:** This work was partially funded by the Labex IGO program supported by the National Research Agency via the investment of the future program ANR-11-LABX-0016-01. This work was supported by an Etoiles Montantes from Pays de la Loire to C.G. This work was also realized in the context of the support provided by the Fondation Progreffe. This project has received funding from the European Union's Horizon 2020 research and innovation program under grant agreement No 825392. The funders had no role in study design, data collection and analysis, decision to publish, or preparation of the manuscript.

**Competing interests:** S.B., I.A. and C.G. have patents on the use of CD8+Treg cells for cell therapy and the diagnosis in immune disorders. The remaining authors declare no competing interests. This does not alter our adherence to PLOS ONE policies on sharing data and materials.

are not efficient at preventing chronic allograft rejection episodes and kidney transplanted patients still lose their graft after some years. Conventional monitoring of kidney transplant patients relies mostly on biological readouts such as creatinine levels, proteinuria or donor-specific antibodies (DSA). Modern risk model or failure scores are trying to combine immune parameters to clinical outcome with some success [2]. In the past 5 years, chemokines, blood transcriptomic biomarkers, donor derived cell-free DNA, donor reactive memory B cells, donor specific CD4$^+$ T cells, CD28$^+$CD4$^+$ T cells and NK cells were reported as biomarkers of kidney transplant outcome [3,4]. Although of great potential, they have not yet reached the standard clinical care. This is due to slow and costly clinical implementation and lack of prospective cross-validation between laboratories and clinical centers. Furthermore, single biomarkers have limited predictive reliability. Thus, there is a need of new biomarkers in transplantation to predict and/or monitor easily graft-directed immune responses and adapt IS treatments to avoid adverse effects. The identification of such tool would represent a major breakthrough in the field.

Development of flow cytometry and bioinformatic analyzes opens up new possibilities to identify a combination of markers reflecting better the overall immune responses. Immunophenotyping of blood cells is non-invasive and can be applied repeatedly before and after transplantation to monitor immune responses. Indeed, it is unlikely that the effector or regulatory events occurring at different stages post-transplantation can be revealed with one biomarker or one combination only, thus investigating the immune situation before and after transplantation appears critical.

CD45RC, one of the splicing variants of the CD45 molecule, is a tyrosine phosphatase that is critical in negatively or positively regulating the TCR signaling [5]. CD45RC is a useful and well-known marker to identify naive precursors of conventional Th1 /TEMRA cells (CD45RC$^{hi}$) and memory/regulatory (CD45RC$^{lo/-}$) cells in both CD4$^+$ and CD8$^+$ T cells. Indeed, we and others have reported that FOXP3$^+$ CD4$^+$ and CD8$^+$ regulatory T cells (Tregs) are contained in the CD45RC$^{lo/-}$ cells in mouse, rat and human [6–9]. We showed in Autoimmune Polyendocrinopathy Candidiasis Ectodermal Dystrophy (APECED) and Duchenne Dystrophy that lesioned organs are infiltrated by CD45RC$^+$ cells and that there are significantly more CD45RC$^{hi}$ T cells in blood in APECED patients vs healthy individuals [10,11]. We also showed that treatment with an anti-CD45RC monoclonal antibody inhibits solid organ transplant rejection, GvHD, Duchenne Dystrophy and APECED [9–12]. It was shown that kidney transplanted patients with more than 54.7% of circulating CD8$^+$CD45RC$^{hi}$ T cells before transplantation had a 6-fold increased risk of acute rejection [13–15], and that there was also a correlation with CD4$^+$CD45RC$^{hi}$ T cells [15].

Here, we performed a fluorescent-based high dimensional immunophenotyping of T cells, on a cohort of 69 kidney transplanted patients either with stable graft function or having experienced acute transplant rejection. The analysis was performed before and one year after transplantation or at the time of rejection. We investigated the potential of markers and cell subsets, and more particularly the relevance of T cell segregation on CD45RC and FOXP3 among other markers to predict transplant outcome.

## Materials and methods

### Blood collection and study approval

Blood samples were collected from kidney transplanted patients at the Centre Hospitalier Universitaire (CHU) of Nantes, France. This retrospective study is based on first transplanted patients selected from the Nantes DIVAT biocollection (www.divat.fr) such that 50% had acute rejection occurring within the first 18 months and 50% did not. Rejection (REJ) was

**Table 1. Clinical characteristics of the cohort of kidney transplanted patients.**

| | Before | | After | | Total |
|---|---|---|---|---|---|
| | STA | REJ | STA | REJ | |
| **Total n of patients** | 23 (48.9%) | 24 (51.1%) | 22 (47.8%) | 24 (52.2%) | 69 patients, 93 samples |
| **Graft type (Kidney, Pancreas)** | 2 KL, 21 K | 2 KL, 22 K | 22 K | 1 KL, 23 K | 5 KL - 88 K (5.3% KL - 94.7% K) |
| **Graft rank (1 - 2)** | 22-1 | 21-3 | 20-2 | 22-2 | 85 - 8 (91.3% - 8.7%) |
| **Donor** | deceased | 23 deceased, 1 living | deceased | deceased | 92 deceased (98.9%), 1 living (1.1%) |
| **IS induction (ATG—Simulect)** | 7-16 | 13-11 | 5-17 | 9-15 | 34-59 (36.5% - 63.4%) |
| **IS treatment (CsA—TAC)** | 2-23 | 4-20 | 0-22 | 1-23 | 7-86 (7.5% - 92.5%) |
| **Follow up (days post transplantation)** | 1720 ± 134.4 | 1576 ± 210 | 1848 ± 113.3 | 1450 ± 153.0 | 1643 ± 79.7 |
| **Back to dialysis** | 2 | 8 | 0 | 3 | 13 (13.9%) |
| **Deceased recipients** | 0 | 2 | 0 | 4 | 6 (6.4%) |
| **Rejection timing (days post transplantation)** | NA | 172.1 ±34.1 | NA | 212.8 ±34.3 | 192.5 ±24.1 |
| **Delay transplantation- blood sample (days post transplantation)** | NA | NA | 375.5 ±2.6 | 212.8 ±34.3 | 288.8 ±21.9 |
| **Delay sample—REJ (days)** | NA | NA | NA | range ±5, mean = 0.08 ± 0.33 | range ±5, mean = 0.08 ± 0.33 |
| **TCMR / ABMR / Fibrosis** | NA | 56.5% / 26% / 43.5% | NA | 45.8% / 12.5% / 50% | 51.2% / 32.3% / 46.8% |
| **Age (year-old)** | 48.8 ±2.7 | 48.3±2.5 | 52.0 ±3.1 | 51.7 ±3.1 | 50.2 ±1.4 |
| **Sex (% F)** | 48% | 54% | 32% | 39% | 42% |
| **Sex combination (R/D: F/F, F/M, M/F, M/M)** | 6,5,8,4 | 6,7,4,7 | 4,3,8,7 | 4,5,3,12 | 20,20,23,30 (21.5%, 21.5%, 24.7%, 32.2%) |

defined by humoral, cellular or borderline rejection according to Banff classification in effect at the time of diagnosis proven by biopsy. None of these signs were observed in systematic biopsy on month+12 in stable (STA) patients. A stable function of the graft is defined as stable creatinine below 150 μmol/L (ideally < 100 μmol/L), zero proteinuria or less than 0.5 g/24h or g/g, an immunosuppressive treatment other than Sirolimus (Rapamune) or Everolimus (Certican), a clearance greater than 40 ml/min in MDRD and no DSA for more than one year. CNI doses were adapted according to the clinical criteria for graft rejection in order to minimize side effects. Therefore, patients defined as STA received lower doses of CNI than REJ patients. Blood was collected before anti-rejection treatment. Blood from healthy individuals was collected at the Etablissement Français du sang (Nantes, France) following signature of a written informed consent form. Blood was collected in EDTA tubes, PBMCs were isolated by Ficoll gradient, frozen in autologous serum supplemented with 10% DMSO, and stored at the Biologic Resources Center of the CHU of Nantes (CRB, BRIF: BB-0033-00040). Cohort characteristics are described in Table 1.

## Flow cytometry on PBMCs

PBMCs were thawed in warm RPMI1640 medium supplemented with HEPES, MEM, sodium pyruvate, glutamine and 10% FCS, and stimulated with 50ng/mL PMA, 5μg/mL ionomycin in the presence of 10μg/mL Brefeldin A for 4h at 37˚C 5% $CO_2$. Cells were washed with PBS 1X, stained with Live/dead Fixable yellow dead cell stain kit (ThermoFisher), Fc receptors were blocked before membrane staining with antibodies and then fixed with 2% paraformaldehyde (PFA). For intracellular staining, the cells were permeabilized 30 min at 4˚C with FOXP3/Transcription Factor Staining Buffer Set (ThermoFisher scientific), stained for 45 min at room

temperature and then fixed with 2% PFA. Monoclonal antibodies used are listed in S1 Table. LSR II (BD Biosciences) and FlowJo Software were used to analyze the cells.

## Statistical analyses

The Mann Whitney test was used to compare cell frequency in STA vs REJ patients and Wilcoxon matched-pairs signed rank test for comparison of patients before vs after transplantation. For the correlation analyzes between incidence of transplant rejection and the frequency of cells, patients were divided on the median cell frequency and the two groups were compared by Log Rank (Mantel Cox) test. Linear regression was used to analyze the correlation of frequency of cell subsets with recipient age or graft survival. Predictive values of subsets frequency were performed with Random Forest algorithm and analyzed using receiver operating characteristics (ROC) curves to assess the specificity and sensitivity of the panel of markers to predict graft outcome. Statistical analysis was performed using GraphPad prism and R. A p-value lower than 0.05 was considered statistically significant.

## Results

### Characteristics of the kidney transplanted patient cohort

69 patients, who underwent kidney transplantation and half of whom rejected their graft in the first 18 months, were selected from the DIVAT biocollection for blood immunophenotyping. The demographic and clinical characteristics of these patients are described in Table 1. Among them, 91.3% were first-rank transplants, 5.3% were kidney-pancreas transplants, 98.9% of the grafts were provided by deceased donors, 36.5% were treated with ATG and 63.4% with Simulect for induction and all were treated with CNIs (cyclosporine or tacrolimus). Patients were monitored for 4.5 years after transplantation, 13.9% returned to dialysis and 6.4% died.

Among the 69 patients, 45 patients were immunophenotyped before or after transplant and 24 patients before and after transplant. Immunophenotype was performed at month +12 for patients having a stable graft function for 18 months (STA), or at the time of rejection (mean ±SEM = 212.8 ± 34.3 days, REJ). The patients were 50.2 ± 1.4 years old, 42% of them were female, 53.8% were matched with the donor for gender, and all were evenly distributed across all groups with no impact on the incidence of transplant rejection (S1A–S1C Fig). There was no correlation between the frequency of CD4+ (S1D Fig) or CD8+ T cells (S1E Fig) in the peripheral blood of patients and the incidence of rejection or the time of immunophenotyping (S1F Fig).

### Higher frequency of CD45RC^lo/- FOXP3+ Tregs in patients with stable graft function before transplantation

Previous studies showed that kidney transplant patients with high frequency of CD8+CD45RC^hi T cells before transplantation had a higher risk of acute rejection [13–15]. To confirm the clinical relevance of CD45RC for monitoring immune response after transplantation, we analyzed its expression on circulating T cells before and after transplantation in the cohort of kidney transplant patients described above (Table 1). First, we observed that the age of patients in the entire cohort had a minor impact on the frequency of CD45RC^hi in CD4+ (Fig 1A) or CD8+ (Fig 1B) T cells since it accounted for only 9.8% of the variation within CD4+ T cells and 11.25% in CD8+ T cells, consistently with previous publication [13]. Similarly, the gender of the recipients had no impact on CD45RC expression in T cell subsets (Fig 1C). In addition, we observed no significant difference in the percentage of CD45RC^hi T cells

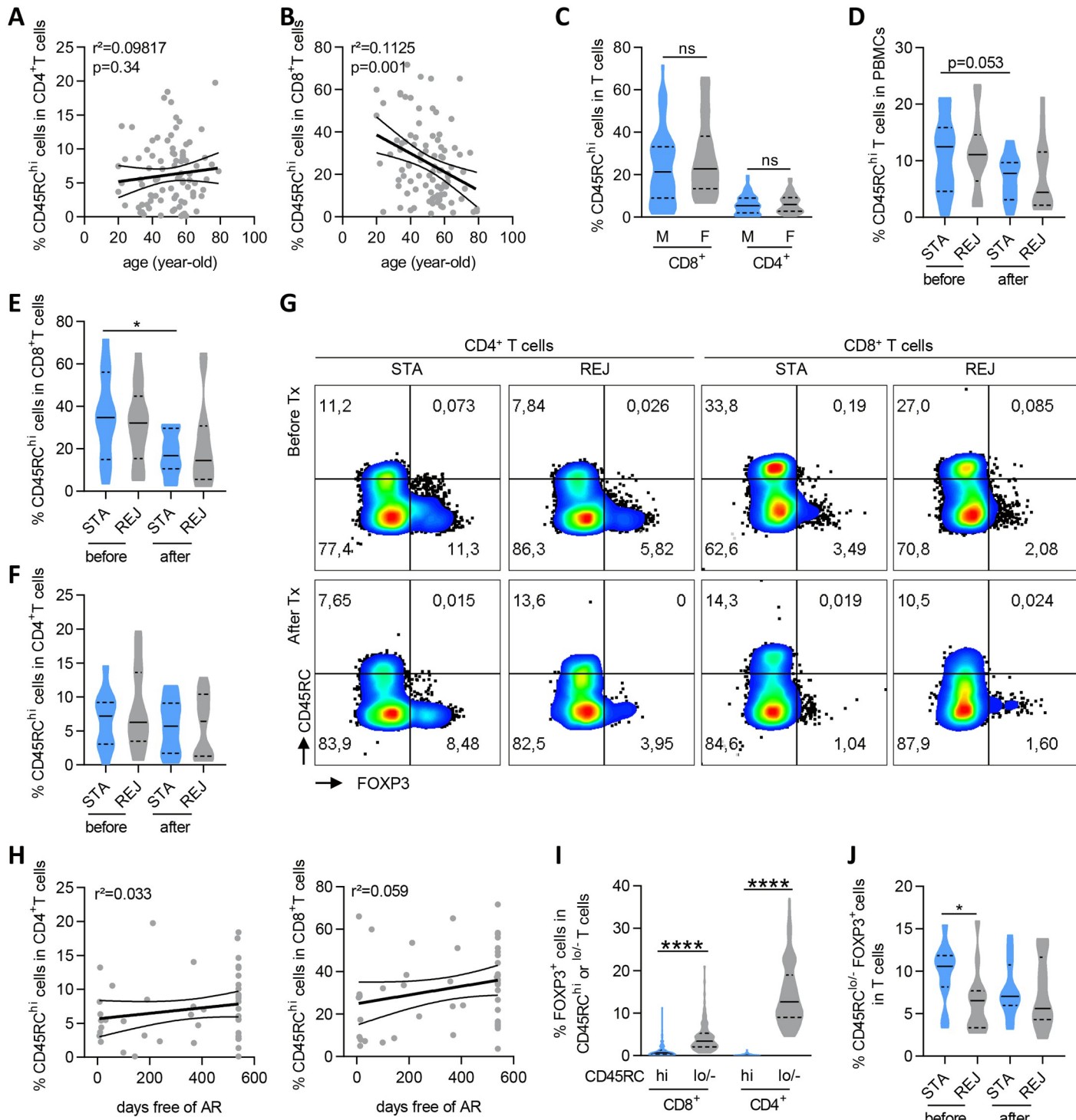

**Fig 1. Higher frequency of CD45RC<sup>lo/-</sup>FOXP3<sup>+</sup> Tregs in patients with stable graft function before transplantation. (A-B)** Correlation analysis of frequency of CD45RC$^{hi}$ cells in CD4$^+$ **(A)** and CD8$^+$ **(B)** T cells with the age of recipients. n = 93 samples. Thick line = linear regression, thin lines = 95% confidence. **(C)** Frequency of CD45RC$^{hi}$ cells in CD8$^+$ and CD4$^+$ T cells in male (M, n = 53) or female (F, n = 40) recipients. **(D-F)** Patients with a stable graft function (STA) or who have experienced transplant rejection episodes (REJ) over 18 months were analyzed before and after transplantation, at the time of rejection or 1 year after kidney transplantation for frequency of CD45RC$^{hi}$ T cells in PBMCs **(D)**, in CD8$^+$ **(E)** and CD4$^+$ **(F)** T cell subsets. **(G)** Representative dot plots of CD45RC and FOXP3 stainings on CD4$^+$ and CD8$^+$ T cells from STA and REJ patients before (upper) and after (lower) transplantation. **(H)** Correlation analysis of the frequency of CD45RC$^{hi}$ cells in CD4$^+$ (left) and CD8$^+$ (right) T cells before transplantation with time free of acute rejection (AR) episodes. n = 47. Thick line = linear regression, thin lines = 95% confidence. **(I)** Frequency of FOXP3$^+$ cells in CD45RC$^{hi}$ or $^{lo/-}$ CD8$^+$ and CD4$^+$ T cells in all patients. n = 93 samples. **(J)** Frequency of CD45RC$^{lo/-}$

FOXP3$^+$ cells in T cells from STA and REJ patients before and after transplantation. STA, n = 11; REJ, n = 13. Wilcoxon matched-pairs signed rank test for comparison of patients over time and CD45RC subsets within T cells; Mann Whitney test for STA vs REJ groups comparison; Linear regression for correlation analyses. *p<0.05, ****p<0.0001. Violin plots: Solid line: Median, dotted lines: Quartiles.

in PBMCs, in CD8$^+$ or in CD4$^+$ T cells between STA and REJ patients, before or after transplantation (Fig 1D–1G). In this cohort, the proportion of CD45RC$^{hi}$ cells in CD4$^+$ and CD8$^+$ T cell subsets before or 1-year post transplantation did not correlate with rejection-free transplant survival (Figs 1H and S1G). However, we observed a significant decrease in the proportion of CD45RC$^{hi}$ cells in CD8$^+$ (Fig 1E and 1G) but not CD4$^+$ T cells (Fig 1F and 1G) in STA patients over time and a trend for REJ patients, probably due to the immunosuppressive regimen treatment following transplantation. Furthermore, FOXP3$^+$ T cells, which are contained in the CD45RC$^{lo/-}$ subsets (Fig 1G and 1I), were significantly more frequent in STA compared to REJ patients before transplantation (Fig 1J).

These results show that the percentage of CD45RC$^{hi}$ in CD8$^+$ T cells is decreased in blood following transplantation, likely due to IS, and does not correlate by itself with graft function. In contrast, the percentage of CD45RC$^{lo/-}$ FOXP3$^+$ T cells is higher in stable patients before transplantation and tend to correlate with better graft function (S1H Fig).

## Stable graft function correlates with a lower expression of CD28, PD-1 and HLA-DR cells in CD45RC$^{lo/-}$ T cell subsets after transplantation

Since CD45RC alone was not sufficiently predictive of graft function, we analyzed the concomitant expression of CD28, PD-1, HLA-DR, IFNγ, GITR, CD154, CD127, CD45RA, CD27, CD103, IL-10 and T-bet by CD45RC$^{hi}$ and CD45RC$^{lo/-}$ FOXP3$^{+/-}$ T cell subsets at 1 year post transplantation in 22 STA patients and at the time of rejection in 24 REJ patients to identify cell subsets correlating with the alloimmune response.

While analysis of IFNγ, GITR, CD154, CD127, CD45RA, CD27, CD103, IL-10 and T-bet expression as single markers among CD45RC T cell subsets did not correlate with incidence of acute rejection, we observed a correlation for the CD28, PD-1 and HLA-DR markers (Fig 2).

Indeed, analysis of CD28 expression, a marker previously associated with CD8$^+$ Treg cells [16], showed that CD28$^-$CD45RC$^{lo/-}$CD8$^+$ T cells expanded significantly more over time in STA than in REJ patients (S2A and S2B Fig). Overall, we observed that patients with more than 50% of CD28$^-$ expression within FOXP3$^-$ and FOXP3$^+$ CD8$^+$ CD45RC$^{lo/-}$T cells showed a 2.15 and 2.43-fold lower incidence of acute rejection over 18 months, respectively (Fig 2A–2C). By contrast, the expression of CD28 in the CD45RC$^{hi}$ compartment was not predictive (S2C and S2D Fig). These results suggest that CD28$^-$CD45RC$^{lo/-}$FOXP3$^+$CD8$^+$ Treg cells are associated with a more stable graft function.

We previously observed a higher expression of PD-1 in the CD45RC$^{lo/-}$ subset vs CD45RC$^{hi}$ CD8$^+$ T cells in healthy subjects, but a similar expression of PD-1 in FOXP3$^+$ and FOXP3$^-$ CD8$^+$CD45RC$^{lo/-}$ T cells [6]. However, we observed a higher frequency of PD-1$^+$ cells in CD8$^+$CD45RC$^{lo/-}$ both FOXP3$^-$ and FOXP3$^+$ in REJ patients, and a significant positive correlation between a high frequency of PD-1$^+$ cells in CD8$^+$CD45RC$^{lo/-}$FOXP3$^+$ T cells and a higher incidence of rejection (Fig 2D–2F). Indeed, patients exhibiting more than 78.45% PD-1$^+$ cells in CD8$^+$CD45RC$^{lo/-}$FOXP3$^+$ T cells have 2-fold higher incidence of acute rejection. These results exclude a role for PD-1 as a biomarker for CD8$^+$ Treg-associated tolerance, but rather as a biomarker of acute rejection.

Analysis of HLA-DR demonstrated a significantly higher frequency of HLA-DR$^+$ in CD8$^+$CD45RC$^{lo/-}$FOXP3$^-$ and FOXP3$^+$ T cells in REJ patients and a high frequency of

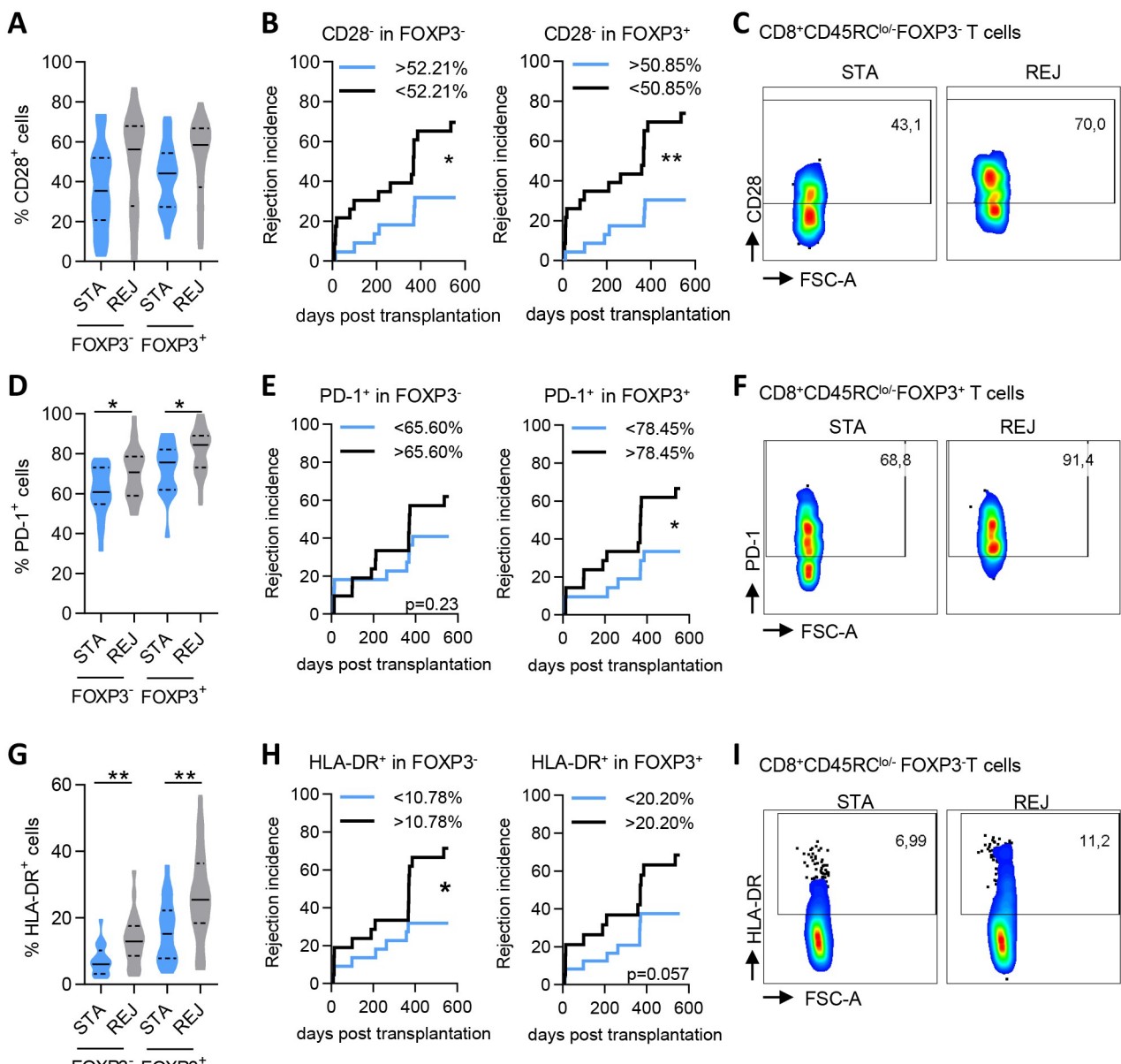

**Fig 2. Stable graft function correlates with a lower expression of CD28, PD-1 and HLA-DR cells in CD45RC$^{lo/-}$ T cell subsets in kidney transplanted patients.** STA and REJ patients were analyzed on month +12 post-transplantation or at the time of rejection respectively. **(A)** Frequency of CD28$^+$ cells in FOXP3$^-$ and FOXP3$^+$ CD8$^+$CD45RC$^{lo/-}$ T cells. STA, n = 22; REJ, n = 24. **(B)** Incidence of allograft rejection for patients having more (blue line) or less (black line) than 52.21% CD28$^-$ cells in CD8$^+$CD45RC$^{lo/-}$FOXP3$^-$ T cells or 50.85% in CD8$^+$CD45RC$^{lo/-}$FOXP3$^+$ T cells. **(C)** Representative dot plots of CD28 staining on CD8$^+$CD45RC$^{lo/-}$ FOXP3$^-$ T cells. **(D)** Frequency of PD-1$^+$ cells in FOXP3$^-$ and FOXP3$^+$ CD8$^+$CD45RC$^{lo/-}$ T cells. STA, n = 20; REJ, n = 23. **(E)** Incidence of allograft rejection for patients having more (black line) or less (blue line) than 65.60% PD-1$^+$ cells in FOXP3$^-$CD8$^+$CD45RC$^{lo/-}$ T cells (left) or 78.45% PD-1$^+$ cells in FOXP3$^+$CD8$^+$CD45RC$^{lo/-}$ T cells (right). **(F)** Representative dot plots of PD-1 staining on CD8$^+$CD45RC$^{lo/-}$ FOXP3$^+$ T cells. **(G)** Frequency of HLA-DR$^+$ cells in FOXP3$^-$ and FOXP3$^+$ CD8$^+$CD45RC$^{lo/-}$ T cells. STA, n = 20; REJ, n = 23. **(H)** Incidence of allograft rejection for patients having more (black line) or less (blue line) than 10.78% HLA-DR$^+$ cells in CD8$^+$CD45RC$^{lo/-}$ FOXP3$^-$ T cells (left) or 20.20% HLA-DR$^+$ cells in CD8$^+$CD45RC$^{lo/-}$ FOXP3$^+$ T cells (right). **(I)** Representative dot plots of HLA-DR staining on CD8$^+$CD45RC$^{lo/-}$ FOXP3$^-$ T cells. Volin plots solid line: Median, dotted lines: Quartiles. Mann Whitney test for STA vs REJ groups comparison; Log-rank (Mantel Cox) test for correlation analyzes between incidence of transplant rejection and the frequency of cells. *p<0.05, **p<0.01.

HLA-DR$^+$CD8$^+$CD45RC$^{lo/-}$FOXP3$^-$ T cells positively correlated with a higher incidence of rejection (Fig 2G–2I). FOXP3$^+$CD45RC$^{lo/-}$ CD4$^+$ Tregs showed higher frequency of HLA-DR$^+$ cells in REJ patients but no significant correlation with rejection incidence (S2E–S2G Fig).

Importantly, we observed no impact of induction treatment such as Simulect or ATG on our findings (S2H–S2L Fig).

Overall, these results support an association between CD28, PD-1 or HLA-DR expression together with CD45RC, FOXP3 and graft function.

## Markers expressed by CD45RC T cell subsets after transplantation reflect graft function

Using ROC curve to determine the diagnostic ability of the expression of the 14 individual markers on the different cell subsets, we were able to go further in the analysis and in addition to the previously evidenced markers, we are also able to demonstrate a negative correlation of CD154 expression on CD8$^+$CD45RC$^{lo/-}$FOXP3$^+$ T cells with the rejection incidence (Fig 3A, AUC = 0.714), evidencing CD154 as an acceptable marker of stable graft function. Although to a lesser extent, we also observed a trend for a correlation of the expression of CD103 and CD45RA on CD8$^+$CD45RC$^{lo/-}$ T cells (AUC = 0.696 and AUC = 0.670, respectively), IL-34 on CD8$^+$CD45RC$^{hi}$ T cells (AUC = 0.652), and CD28 and TGFβ on CD4$^+$CD45RC$^{lo/-}$ T cells (AUC = 0.652 and AUC = 0.661, respectively), CD28 and PD-1 on CD4$^+$CD45RC$^{lo/-}$FOXP3$^+$ T cells (AUC = 0.670 and AUC = 0.652, respectively) and GITR on CD4$^+$CD45RC$^{hi}$ T cells (AUC = 0.661) with graft survival free of acute rejection (Fig 3B).

Then, we constructed multi-parameter ROC curves using the expression of several markers by T cell subsets to build a new predictive model of the transplant outcome. The selection of markers whose individual expression correlated with the incidence of transplant rejection, such as CD28, PD-1, HLA-DR, CD103, CD154, GITR, and IFNγ, on FOXP3$^+$CD45RC$^{lo/-}$ CD4$^+$ and CD8$^+$ Tregs had limited predictive capacity (AUC = 0.67) (S2M Fig). In contrast, the selection of markers and T cell subsets that individually tended to correlate with the outcome of the transplant significantly improved the prediction model (AUC = 0.71) (Fig 3B and 3C) and finally we identified markers that could help predicting a stable graft function: CD103 (with a median expression in all patients >1.03%), CD45RA (>64.65%), HLA-DR (<10.78%) and CD28 (<47.79%) on CD8$^+$CD45RC$^{lo/-}$FOXP3$^-$ T cells; CD154 (>5.42%), CD28 (<49.15%), and PD-1 (<78.45%) in CD8$^+$CD45RC$^{lo/-}$FOXP3$^+$ T cells; IL-34 (<22.65%) in CD8$^+$CD45RC$^{hi}$ T cells; CD28 (>66.45%) and TGFβ (<35.0%) in CD4$^+$CD45RC$^{lo/-}$FOXP3$^-$ T cells; CD28 (<52.3%) and PD-1 (<71.2%) in CD4$^+$CD45RC$^{lo/-}$FOXP3$^+$ Tregs; GITR (>3.25%) in CD4$^+$CD45RC$^{hi}$ T cells.

Altogether we generated 2 ROC curves using different parameters and predictive of graft outcome.

## PD-1, GITR, IL-34 and CD127 expression in CD45RC T cell subsets are predictive of graft rejection before transplantation

Predicting transplant outcome based on PBMCs analysis before organ transplantation or as early as possible after transplantation is a major objective to personalize immunosuppressive treatments. First, we analyzed the expression of each of the 14 selected markers on circulating CD4$^+$ and CD8$^+$ CD45RC T cell subsets before transplantation. Interestingly, we observed that patients exhibiting more than 18.50% of expression of PD-1 in CD8$^+$CD45RC$^{hi}$ T cells had a 1.85-fold higher risk of graft rejection incidence (Fig 4A). Importantly, PD-1 expression on total CD4$^+$ and CD8$^+$ T cells were not predictive of the graft outcome (S3A and S3B Fig). In addition, patients exhibiting more than 23.60% of expression of GITR in

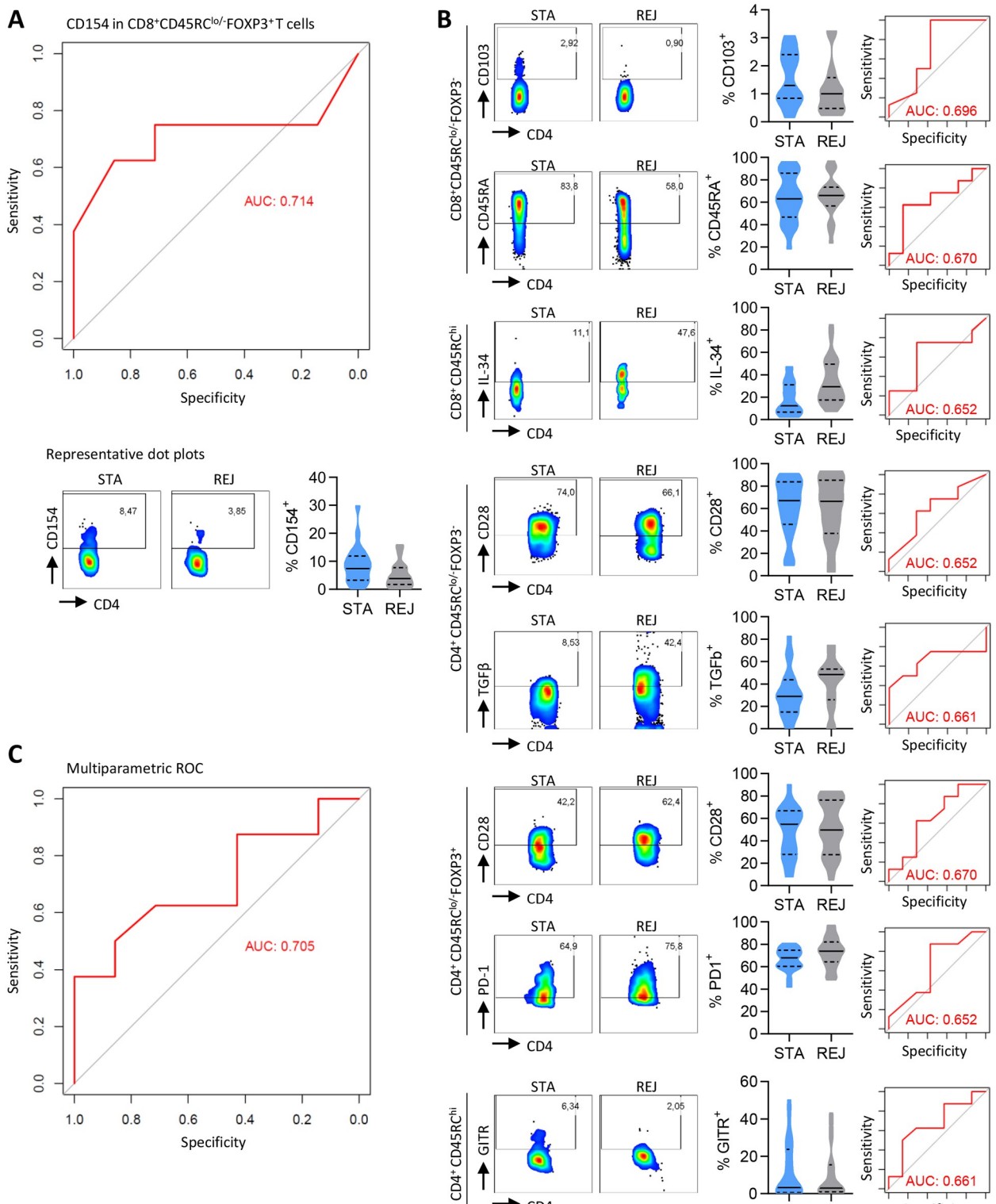

**Fig 3. Markers expressed by CD45RC T cell subsets in transplanted patients are biomarkers of graft function. (A) Upper**: ROC curves illustrating the specificity and sensitivity of the diagnosis of rejection based on the expression of CD154 in CD8+CD45RClo/-FOXP3+ T cells. **Lower**: Representative dot plots and violin plots of the expression of CD154 in CD8+CD45RClo/-FOXP3+ T cells in STA and REJ patients. **(B)** Representative dot plots and violin plots in STA and REJ patients and ROC curves illustrating the specificity and sensitivity of the diagnosis of rejection based on expression of CD103 and CD45RA in CD8+CD45RClo/-FOXP3- T cells, IL-34 in CD8+CD45RChi T cells, CD28 and TGFβ in

CD4$^+$CD45RC$^{lo/-}$ FOXP3$^-$ T cells, CD28 and PD-1 in CD4$^+$CD45RC$^{lo/-}$FOXP3$^+$ T cells and GITR in CD4$^+$CD45RC$^{hi}$ T cells. Solid line: Median, dotted lines: Quartiles. Mann Whitney test, ns. (**C**) ROC curves illustrating the specificity and sensitivity of the diagnosis of rejection based on the expression of markers in (**A**) and (**B**).

CD8$^+$CD45RC$^{lo/-}$FOXP3$^-$ T cells had a 1.75-fold higher risk of graft rejection incidence (Fig 4B). By contrast, patients exhibiting more than 6.60% of expression of IL-34 in CD4$^+$CD45RC$^{hi}$ T cells had a 1.75-fold lower risk of graft rejection incidence (Fig 4C). Using a single parameter ROC curve analysis, we observed that the expression of CD127 on CD4$^+$CD45RC$^{lo/-}$FOXP3$^+$ T cells was predictive of graft rejection (AUC = 0.745, Fig 4D). None of the other markers was considered reliable enough to predict the graft outcome. Similarly, a multiparametric ROC curve analysis method was not strong enough to predict the transplant outcome before transplantation in our cohort of patient.

## Identification of markers expressed by CD45RC T cell subsets evolving with the immune response to monitor transplant outcome

To assess the feasibility of monitoring patient transplant by phenotyping blood cells and adapting immunosuppressive treatments in real time, we performed a longitudinal analysis of CD45RC T cell subsets phenotype, in patients related before transplantation and after transplantation.

In STA patients, the percentage of CD28$^+$ cells in both CD45RC$^{hi}$ CD8$^+$ and CD4$^+$ T cells, likely naive cells, dropped significantly (Fig 5A), while the frequency of CD28$^-$ cells in CD8$^+$CD45RC$^{lo/-}$ T cells increased (S2A Fig). Consistently with previous studies [17], these results suggest that Treg-like cells, may emerge to control the immune response.

In accordance with the association of PD-1 expression with graft rejection (Fig 2D–2F), PD-1$^+$ cells frequency was significantly increased in CD4$^+$CD45RC$^{lo/-}$ T cells, and to a lower extent in CD8$^+$CD45RC$^{lo/-}$ T cells, in REJ, but not STA patients (Fig 5B). Similarly, HLA-DR$^+$ cells were significantly more frequent in CD8$^+$ and CD4$^+$CD45RC$^{lo/-}$ T cells in REJ, but not in STA patients (Fig 5C).

In addition, the increased frequency of PD-1$^+$ and HLA-DR$^+$ CD8$^+$CD45RC$^{hi}$ T cells in REJ patients and decreased frequency of CD127$^+$ CD4$^+$ and CD8$^+$CD45RC$^{hi}$ T cells and T-bet$^+$ CD8$^+$CD45RC$^{hi}$ T cells in STA patients (Fig 5D and 5E) may reflect the inflammatory response overtime. Finally, we observed a trend for an upregulation of IFN$\gamma$ in CD4$^+$CD45RC$^{hi}$ T cells in REJ patients (Fig 5F).

Altogether, we identified subsets of activated Tconv cells expanded in REJ patients (HLA-DR$^+$, PD-1$^+$, IFN$\gamma^+$) or decreased in STA patients (CD127$^+$, T-bet$^+$) and subsets of Tregs promoted in STA patients (CD28$^-$) which should be considered to monitor graft outcome.

## Discussion

Kidney transplantation is the best therapy for patients with end-stage kidney diseases. Currently in clinical care post kidney transplantation, physicians rely on serum creatinine/eGFR, proteinuria, hematuria, sign of intravascular hemolysis and BK virus viral load to adapt their management. Although these biological markers are non-invasive, they lack sensitivity and specificity [18–20]. In addition, immunosuppressive drugs have been very successful in the control of acute rejection but do not inhibit for some patients graft rejection. Thus, improvement of surveillance/prediction of allograft dysfunction based on immunological markers in addition to the biological markers would improve survival and quality of life, even allowing for

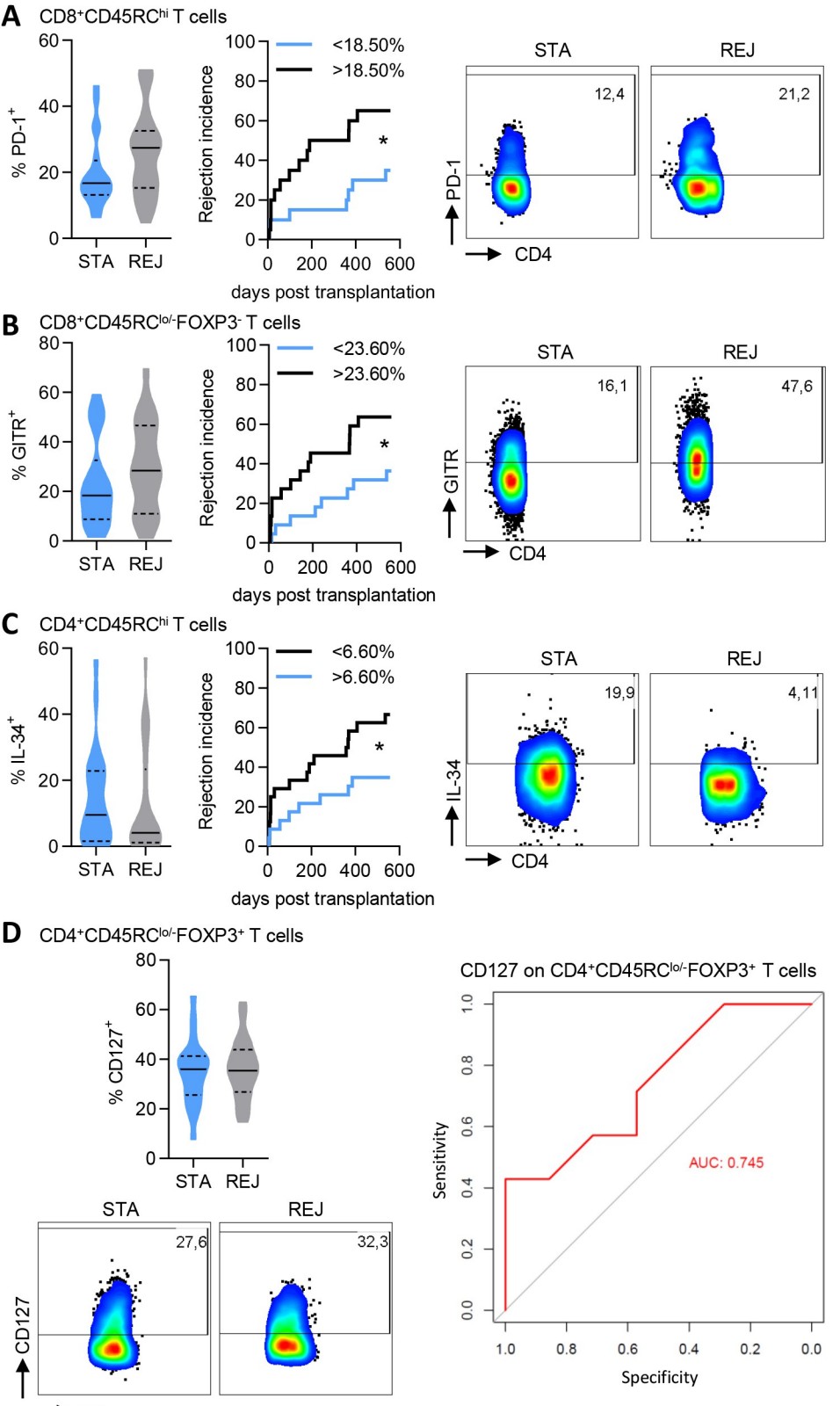

**Fig 4. PD-1, GITR, IL-34 and CD127 expression in CD45RC T cell subsets are predictive of graft rejection before transplantation.** STA and REJ patients were analyzed before transplantation. **(A) Left:** Frequency of PD-1⁺ cells in

CD8+CD45RC^hi T cells. **Middle:** Incidence of allograft rejection for patients having more (black line) or less (blue line) than 18.50% PD-1+ cells in CD8+CD45RC^hi T cells. STA, n = 20; REJ, n = 20. **Right:** Representative dot plots of PD-1 staining on CD8+CD45RC^hi T cells. **(B) Left:** Frequency of GITR+ cells in CD8+CD45RC^lo/- FOXP3- T cells. **Middle:** Incidence of allograft rejection for patients having more (black line) or less (blue line) than 23.60% GITR+ cells in CD8+CD45RC^lo/- FOXP3- T cells. STA, n = 22; REJ, n = 22. **Right:** Representative dot plots of GITR staining on CD8+CD45RC^lo/- FOXP3- T cells. **(C) Left:** Frequency of IL-34+ cells in CD4+CD45RC^hi T cells. **Middle:** Incidence of allograft rejection for patients having more (blue line) or less (black line) than 6.60% IL-34+ cells in CD4+CD45RC^hi T cells. STA, n = 23; REJ, n = 24. **Right:** Representative dot plots of IL-34 staining on CD4+CD45RC^hi T cells. **(D) Upper left:** Frequency of CD127+ cells in CD4+CD45RC^lo/- FOXP3+ T cells. STA, n = 20; REJ, n = 21. **Bottom left:** Representative dot plots of CD127 staining on CD4+CD45RC^lo/- FOXP3+ T cells. **Right:** ROC curves illustrating the specificity and sensitivity of the prognosis of graft rejection based on the expression of CD127 in CD4+CD45RC^lo/- FOXP3+ T cells. Violin plots solid line: Median, dotted lines: Quartiles. Mann Whitney test for STA vs REJ groups comparison, ns. Log-rank (Mantel Cox) test for correlation analyzes between incidence of transplant rejection and the frequency of cells, *p<0.05.

optimal potentially weaned post-transplantation treatment. The ability to stratify patients using a marker or a combination of markers for an increased risk of transplantation rejection is still a major challenge and most biomarkers have failed to do so in the past decades and have not been implemented as standard of care to enhance post-transplant follow-up [3].

In the present work, using multicolor immunophenotyping of T cells from PBMCs in a cohort of 69 patients with either stable graft function or having experienced acute transplant rejection analyzed in a retrospective manner before transplantation and one year after transplantation or at the time of rejection, we identified combinations of markers and cell subsets

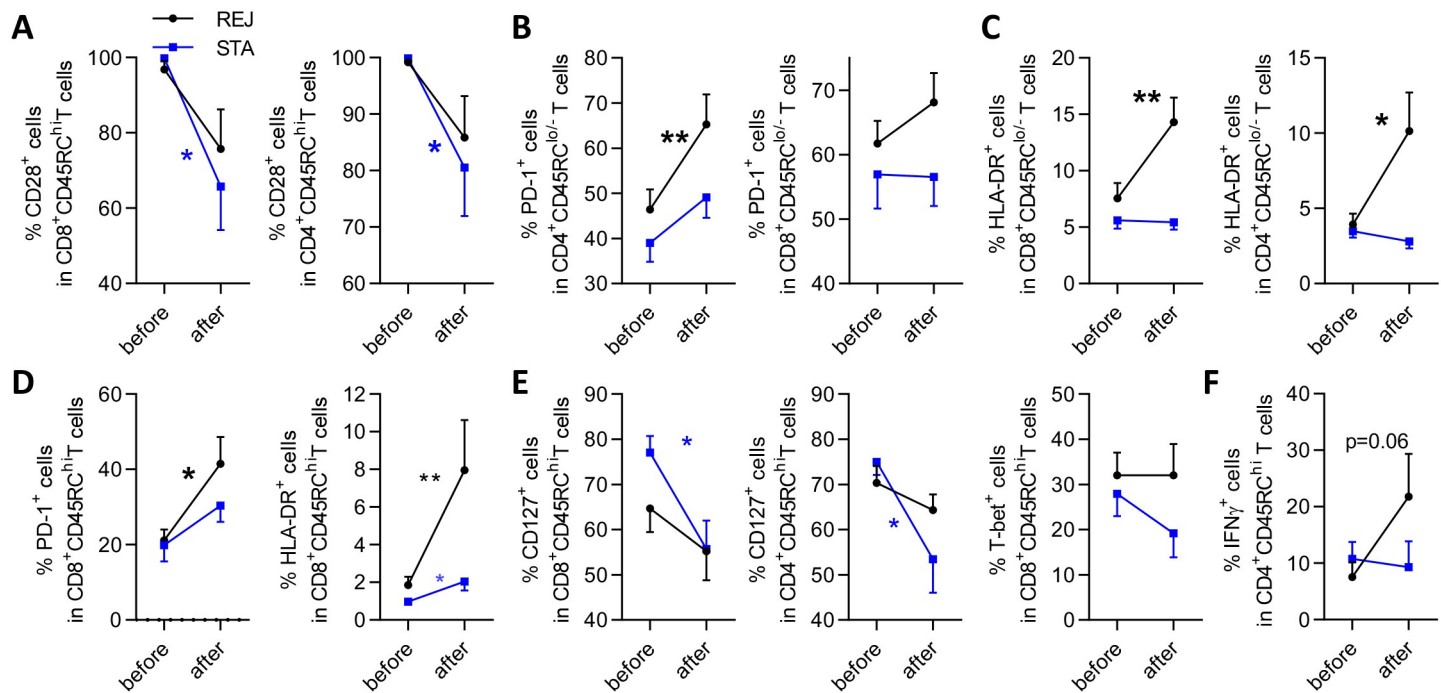

**Fig 5. Identification of markers expressed by CD45RC T cell subsets evolving with the immune response to monitor graft outcome.** 11 STA (blue squares) or 13 REJ (black circles) patients were compared for evolution of marker expression in CD45RC T cell subsets before and after transplantation. Mean +/- SEM of frequency of CD28+ (**A**), PD-1+ (**B, D**), HLA-DR+ (**C-D**), CD127+ and T-bet+ (**E**), IFNγ+ (**F**) cells in CD8+CD45RC^hi (**A** left, **D, E** left and right), CD8+CD45RC^lo/- (**B** right, **C** left), CD4+CD45RC^hi (**A** right, **E** middle, **F**) or CD4+CD45RC^lo/- (**B** left, **C** right) T cells. Wilcoxon matched-pairs signed rank test for before vs after transplantation comparison, *p<0.05, **p<0.01, black * for REJ patients, blue * for STA patients.

as potential biomarkers of transplant outcome, and explored the importance of cell segregation on the CD45RC and FOXP3 markers to identify the signature of stable graft function.

Three prospective studies demonstrated that CD45RC expression in T cells, but not FOXP3[+] T cells, from PBMCs before transplantation predicted acute rejection in cohort studies of 89 and 128 kidney transplant patients [13–15]. They showed that pre-transplant proportion of CD8[+]CD45RC[hi] T cells is associated with a 4 to 6-fold increased risk of acute rejection, mainly T cell mediated acute rejection. This increased expression of CD45RC on CD8[+] T cells was also described on PBMCs from lupus nephritis patients compared to healthy volunteers, although it was not analyzed whether increased expression correlated with poorer outcome [7] and on PBMCs from a cohort of APECED patients [11]. However, in our cohort of patients, the expression of CD45RC on T cells was not sufficient for prognosis of transplant rejection per se. This may be due to the smaller cohort of patients compared to what was used previously by other groups. However, distinction of conventional and regulatory T cells using the CD45RC and FOXP3 markers was required to highlight correlation of T cell subsets with the immune status of the patients. Targeting CD45RC with a depleting mAb showed high efficacy to establish tolerance in models of heart transplantation, GvHD, APECED and Duchenne's dystrophy [9–12]. The short-term effect was mediated by pathogenic cell depletion, while the long-term therapeutic effect in these models was mediated by Tregs of increased suppressive activity able to induce infectious tolerance. Indeed, since CD45RC is the only CD45 isoform not expressed by Tregs, it represents a promising therapeutic to target non Tregs in transplant patients.

The expression of the costimulatory receptor CD28 is usually used to identify naive and central memory (CM) cells (CD28[+]) from effector memory (EM) and terminally differentiated (TEMRA) cells (CD28[-]) in CD4[+] and CD8[+] T cells [21], and to identify a subset of CD8[+] Tregs [16]. Interestingly, CD28[-]CD8[+] T cells were reported to be expanded in patients free of acute rejection of heart transplantation [17] and associated to a lower risk for late rejection of kidney transplantation [22] but also associated with late graft dysfunction [23]. Indeed, we observed that, except for the CD28 marker that was associated with a 2-fold higher incidence of rejection for patients having more than 51.4% CD28[+] cells in CD8[+] T cells consistently with the literature [22], none of the other 13 markers we analyzed on total CD4[+] and CD8[+] T cells individually was sufficient to predict the outcome of the transplant. On CD4[+] T cells, the expression of CD28 was associated to acute cellular rejection in liver or kidney transplanted patients, and can stratify patients at higher risk [24]. In another study, increased pre-transplant frequency of CD28[+]CD4[+] effector memory T cells predicted belatacept-resistant rejection in human renal transplant recipients [25]. Interestingly, the reliability of CD28 as predictive biomarker was maintained when restricting its analysis to CD45RC[lo/-] subset, and its combination with CD45RC and FOXP3 markers even refined the accuracy. However, in healthy volunteers, we have not observed higher suppressive potency in CD28[-] vs CD28[+] in CD8[+]CD45RC[lo/-] Tregs [6,26,27].

PD-1 and HLA-DR expression on CD4[+] and CD8[+] total T cells did not correlate with graft outcome in this cohort either, but their high expression on CD8[+]CD45RC[lo/-]FOXP3[+] and FOXP3[-] Tregs respectively after transplantation and of PD-1 on CD8[+]CD45RC[hi] T cells before transplantation positively correlated with a poorer outcome. The role of PD-1 for Tregs is controversial. PD-1 has been reported being involved in CD4[+]Tregs generation and suppressive properties [28], to identify regulatory cells versus memory cells among CD8[+] Tregs [29], and notably PD-1[+]CD122[+]CD8[+] Tregs have been shown to prolong murine allograft survival [30]. By contrast, other studies reported that PD-1 deficient CD4[+] Tregs were more functional as inhibiting efficiently experimental autoimmune encephalomyelitis or diabetes in mice [31]. Similarly, HLA-DR expression has been controversially associated with T cell activation or the

function of mature CD4[+] Tregs and CD8[+] Tregs [32]. HLA-DR expression has been shown in blood CD8[+] Tregs, expressing a similar phenotype to CD4[+]FOXP3[+] Tregs and suppressing immune responses through cell contact and co-inhibitory molecules [32] and, accordingly, urinary soluble HLA-DR protein has been shown to be a potential biomarker of acute renal rejection [33]. Interestingly, PD-1 was also reported being expressed by exhausted human CD8[+]HLA-DR[+] Tregs [32] and dysfunctional CD4[+] Tregs [34]. The high frequency of HLA-DR[+] cells within CD8[+] and CD4[+] Tregs we observed in the cohort after transplantation correlating with graft rejection supports the hypothesis of exhausted unfunctional Tregs in rejecting patients. Anti-PD-1 mAb treatment is evaluated in patients to stimulate activation of tumor reactive T cells, however it also stimulates PD1[+] CD4[+] and CD8[+] Tregs resulting in higher Treg suppressive activity and unwanted progression of cancer [35].

While anti-PD-1 mAbs might be beneficial for transplanted patients at one year post trans-plant to stimulate Tregs, the high frequency of activated PD-1[+] (and HLA-DR[+]) cells in CD8[+]CD45RC[hi] T cells before the transplantation correlating with poor graft outcome, argues for a later application.

CD4[+]FOXP3[+]Tregs are characterized by a low expression of CD127 as IL-7 is not crucial for Treg survival. However, CD127 expression play a role in Treg homeostasis and can be upregulated by activation [36].

GITR has been reported as a co-activation molecule for conventional T cells and inhibitory molecule for Tregs [37,38]. We recently showed a higher suppressive function by GITR[+]CD45RC[lo/-] CD8[+] T cells and a correlation of GITR expression with FOXP3 expression [6]. Surprisingly, high expression of GITR in CD8[+]CD45RC[lo/-]FOXP3[-] T cells positively corre-lated with higher rejection incidence, suggesting a dual role of GITR for FOXP3[+] and FOXP3[-] CD45RC[lo/-] T cell subsets.

IL-34 was controversially reported associated with chronic inflammation, acute kidney injury and tolerance in transplantation when expressed by CD45RC[lo/-] T cells [39–43]. Indeed, although low, its expression in CD4[+]CD45RC[hi] T cells before transplantation was yet associ-ated with lower rejection incidence.

CD103, the α chain of the αEβ7 integrin, is a specific marker of mouse and human CD4[+] Tregs [44,45], of mouse CD8[+] Tregs inhibiting graft-versus-host disease [46] and of human alloantigen-induced CD8[+] Tregs [26,47] but also, at the same time, a marker of effector cyto-toxic CD8[+] T cells, promoting their migration intragraft into epithelial compartment and involved in allograft rejection in a mouse model of islet allograft [48]. Indeed, CD103 is a mol-ecule rather involved in migration but not in the function of T cell subsets. Thus, its expression by regulatory CD45RC[lo/-] T cells would also promote their migration to the graft to stabilize graft function.

Finally, we were also interested in understanding the evolution of subsets with time. We observed an enrichment in CD28[-] regulatory cells within the CD8[+]CD45RC[lo/-] subset and a decrease in CD28[+] cells in CD4[+] and CD8[+] CD45RC[hi] T cells, ie TEMRA cells, in stable patients, by contrast to an enrichment in PD-1[+] and HLA-DR[+] exhausted cells within CD4[+] and CD8[+] CD45RC[lo/-] T cells in rejecting patients overtime.

The immune response is a complex balance between Tregs and Teffs and considering single markers to predict the graft outcome is utopian. Indeed, these results show that there is no dichotomous marker making it possible to predict the outcome of the transplant but a combi-nation of markers defining the state of activation of cell subsets providing a probability on the status of the graft and of the outcome of the transplant. Some published prediction models include both clinic parameters characteristic and immune profile [23], and we chose to focus rather on the immune cells that are responsible for graft rejection while excluding downstream clinical features. Globally, only a few subsets were correlated with graft outcome when

analyzed before the transplantation and relevant graft-directed subsets were highlighted by the exacerbated immune response toward the graft. Some markers are already clinically investigated and we propose adding at least the CD45RC marker, or even FOXP3, to add value to each of the validated markers without disrupting all the protocols.

We focused our efforts on the investigation of the Teff/Treg balance and profile through broad immunophenotyping however applied to a limited number of patient samples, and were unable to increase the resolution of the prediction to the type of cellular or humoral rejection within the REJ group. New technologies are developing to overview the immune landscape, such as 3'DGE sequencing and single-cell RNA sequencing, however these technologies are limited to mRNA therefore regarding the alternative splicing of markers such as CD45RC, and despite the big data generated, the need for clinicians is a conclusive 1-day experiment. The spectral flow cytometry technology opens a new era of accessible, easy-to-use tool for monitoring clinical platforms. Recently developed multiproteomic analysis of pan T lymphocytes in tissue biopsies makes it possible to deeply characterize the T cell infiltrate of the graft. Our efforts will now focus on finding correlations between graft-infiltrating and blood T cells profile [49] in order to find more precise and relevant blood diagnostic markers of the response to the allograft. In addition, a better understanding of the kinetics of response to allograft by CD8[+] T cell subpopulations will allow real-time diagnosis.

This study reveals the potential of subdividing T cells on CD45RC expression to identify new biomarkers of kidney transplant patient response to treatment and opens new avenues for developing new immunotherapies. The study also highlights the potential of multi-parameter flow cytometry and benefit of implementation in the routine practice.

## Supporting information

**S1 Fig. Characteristics of the transplanted patient cohort. (A)** Violin plots showing the age of STA and REJ recipients of kidney transplantation included in the cohort analyzed before or after transplantation. Solid line: median, dotted lines: quartiles. **(B)** Incidence of transplant rejection in the younger half (black line) or older half (blue line) of patients. Median = 52 year-old. **(C)** Incidence of graft rejection in recipients grafted with sex-matched or mismatched donor. F = female, M = male. **(D)** Incidence of graft rejection in patients that display more (blue line) or less (black line) than 54.30% or 39.10% CD4[+] T cells in PBMCs before (left) or after (right) transplantation respectively. **(E)** Incidence of graft rejection in patients that display more (blue line) or less (black line) than 20.70% or 26.20% CD8[+] T cells in PBMCs before (left) or after (right) transplantation respectively. **(C-E)** n = 45; Log Rank (Mantel Cox) test, ns. **(F)** Correlation analysis of the frequency of CD4[+] (left) and CD8[+] (right) T cells with time post-transplantation. n = 93 samples. **(G)** Correlation analysis of the frequency of CD45RC[hi] cells in CD4[+] (left) and CD8[+] (right) T cells after transplantation with time post-transplantation free of acute rejection (AR) episodes. n = 46. **(F-G)** Thick line = linear regression, thin lines = 95% confidence. **(H)** Incidence of graft rejection in patients that display more (blue line) or less (black line) than 7.7% CD45RC[lo/-]FOXP3[+] cells in T cells before transplantation. Log Rank (Mantel Cox) test, ns.
(PDF)

**S2 Fig. Correlation of CD28, PD-1 and HLA-DR expression in CD8[+]CD45RC[lo/-] T cell subsets and IS treatment with graft outcome. (A-B)** Frequency of CD28[-] cells in CD8[+]CD45RC[lo/-] FOXP3[-] **(A)** and FOXP3[+] **(B)** T cells in STA (n = 11) and REJ (n = 13) patients before and after transplantation. Wilcoxon matched-pairs signed rank test for time comparison and Mann Whitney test for groups comparison, *p<0.05. **(C)** Frequency of CD28[-] cells in CD8[+]CD45RC[hi] T cells. STA, n = 22; REJ, n = 24. **(D)** Incidence of allograft

rejection for patients having more (blue line) or less (black line) than 0.3% CD28$^-$ cells in CD8$^+$CD45RC$^{hi}$ T cells. **(E)** Frequency of HLA-DR$^+$ cells in FOXP3$^-$ and FOXP3$^+$ CD4$^+$CD45RC$^{lo/-}$ T cells. STA, n = 20; REJ, n = 23. Mann Whitney test, **p<0.01. **(F-G)** Incidence of allograft rejection for patients having more (black line) or less (blue line) than 4.98% HLA-DR$^+$ cells in CD4$^+$CD45RC$^{lo/-}$ FOXP3$^-$ T cells **(F)** or 20% HLA-DR$^+$ cells in CD4$^+$CD45RC$^{lo/-}$ FOXP3$^+$ T cells **(G)**. **(H)** Incidence of allograft rejection for patients treated with Simulect (n = 32) or ATG (n = 14) as induction. **(I-L)** Incidence of allograft rejection for patients treated with Simulect and having more (blue line) or less (black line) than 52.21% CD28$^-$ cells in CD8$^+$CD45RC$^{lo/-}$FOXP3$^-$ T cells (**I,** n = 17 STA and 15 REJ), 50.85% CD28$^-$ cells in CD8$^+$CD45RC$^{lo/-}$ FOXP3$^+$ T cells (**J,** n = 17 STA and 15 REJ), 78.45% PD-1$^+$ cells in FOXP3$^+$CD8$^+$CD45RC$^{lo/-}$ T cells (**K,** n = 16 STA and 13 REJ), and 10.78% HLA-DR$^+$ cells in CD8$^+$CD45RC$^{lo/-}$ FOXP3$^-$ T cells (**L,** n = 16 STA and 13 REJ) after transplantation. **(D, F-L)** Log-rank (Mantel Cox) test, **p<0.01, ***p<0.001. **(M)** ROC curves illustrating the specificity and sensitivity of the diagnosis of rejection based on the expression of CD28, PD-1, HLA-DR, CD103, CD154, GITR, and IFNγ, on FOXP3$^+$CD45RC$^{lo/-}$CD4$^+$ and CD8$^+$ Tregs. (PDF)

**S3 Fig. PD-1 expression in CD4$^+$ and CD8$^+$ T cells before transplantation is not predictive of graft outcome.** Incidence of graft rejection in patients that display more (black line) or less (blue line) than 40.29% or 47.63% PD-1$^+$ cells in CD4$^+$ **(A)** or CD8$^+$ **(B)** T cells before transplantation. n = 47. Log Rank (Mantel Cox) test, ns. (PDF)

**S1 Table. mAbs used for immunophenotyping of patients by flow cytometry.** (PDF)

**S1 Graphical abstract.** (PDF)

## Acknowledgments

The authors would like to thank the members of the DIVAT consortium for their involvement in the study, the physicians who helped recruit patients, and all patients who participated in this study. We also thank the clinical research associates who participated in the data collection and investigation, notably C. Kerleau, K. Trébern-Launey and T. Goronflot. Data were collected from the French DIVAT multicentric prospective cohort of kidney and/or pancreatic transplant recipients (www.divat.fr, N° CNIL 914184, ClinicalTrials.gov recording: NCT02900040). The analysis and interpretation of these data are the responsibility of the authors. We thank the biological resource center for biobanking (CHU Nantes, Nantes Université, Centre de ressources biologiques (BB-0033-00040), F-44000 Nantes, France). We thank L. Delbos for her advices in flow cytometry.

## Author Contributions

**Conceptualization:** Séverine Bézie, Ignacio Anegon, Carole Guillonneau.

**Data curation:** Séverine Bézie, Elodie Autrusseau, Nadège Vimond.

**Formal analysis:** Séverine Bézie, Céline Sérazin.

**Funding acquisition:** Carole Guillonneau.

**Methodology:** Séverine Bézie, Ignacio Anegon.

**Project administration:** Carole Guillonneau.

**Resources:** Magali Giral.

**Supervision:** Carole Guillonneau.

**Validation:** Séverine Bézie, Carole Guillonneau.

**Visualization:** Séverine Bézie.

**Writing – original draft:** Séverine Bézie, Carole Guillonneau.

**Writing – review & editing:** Séverine Bézie, Ignacio Anegon, Carole Guillonneau.

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
