## [Decision Letter · Decision Letter 0]

20 Nov 2023

PONE-D-23-20905Renal graft function in transplanted patients correlates with CD45RC T cell phenotypic signaturePLOS ONE

Dear Dr. Guillonneau,

Thank you for submitting your manuscript to PLOS ONE. After careful consideration, we feel that it has merit but does not fully meet PLOS ONE’s publication criteria as it currently stands. Therefore, we invite you to submit a revised version of the manuscript that addresses the points raised during the review process.

We look forward to receiving your revised manuscript.

Kind regards,

Senthilnathan Palaniyandi, Ph.D

Academic Editor

PLOS ONE

Journal Requirements:

"This work was partially funded by the Labex IGO program supported by the National Research Agency via the investment of the future program ANR-11-LABX-0016-01. This work was supported by an Etoiles Montantes from Pays de la Loire to C.G. This work was also realized in the context of the support provided by the Fondation Progreffe. This project has received funding from the European Union’s Horizon 2020 research and innovation program under grant agreement No 825392."

4. We note that you have a patent relating to material pertinent to this article:

"S.B., I.A. and C.G. have patents on the use of CD8+Treg cells for cell therapy and the diagnosis in immune disorders. The remaining authors declare no competing interests."

Please provide an amended statement of Competing Interests to declare this patent (with details including name and number), along with any other relevant declarations relating to employment, consultancy, patents, products in development or modified products etc. Please confirm that this does not alter your adherence to all PLOS ONE policies on sharing data and materials, as detailed online in our guide for authors http://journals.plos.org/plosone/s/competing-interests by including the following statement: ""This does not alter our adherence to  PLOS ONE policies on sharing data and materials.” If there are restrictions on sharing of data and/or materials, please state these. 

Please note that we cannot proceed with consideration of your article until this information has been declared.

7. Your ethics statement should only appear in the Methods section of your manuscript. If your ethics statement is written in any section besides the Methods, please delete it from any other section. 

8. We notice that your supplementary figures and tables are included in the manuscript file. Please remove them and upload them with the file type 'Supporting Information'. Please ensure that each Supporting Information file has a legend listed in the manuscript after the references list.

**Additional Editor Comments:**

We have now received comments from the reviewers of your manuscript, we invite you to submit a revised version of the manuscript. 

Reviewers' comments:

Reviewer's Responses to Questions

**Comments to the Author**

1. Is the manuscript technically sound, and do the data support the conclusions?

Reviewer #1: Yes

Reviewer #2: Yes

2. Has the statistical analysis been performed appropriately and rigorously? 

Reviewer #1: Yes

Reviewer #2: Yes

3. Have the authors made all data underlying the findings in their manuscript fully available?

Reviewer #1: Yes

Reviewer #2: Yes

4. Is the manuscript presented in an intelligible fashion and written in standard English?

Reviewer #1: Yes

Reviewer #2: Yes

5. Review Comments to the Author

Reviewer #1: An illustrative self-explanatory figure that represents the study participants and groups (STA vs. REJ) and the distribution of expression patterns will add to the readability of the manuscript.

It would be better to elaborate more on the correlation of the study finding with the hypothesis of exhausted and unfunctional T regs in rejection. How HLA-DR and PD1 have such a differential expression before and after transplant?

Reviewer #2: The manuscript, Bézie et al., report a CD45RC T cell phenotypic signature associated with the outcome of kidney transplanted patients. This study implies the importance of biomarkers to monitor the transplant outcomes in study participant groups (STA vs. REJ). The results are very interesting, and the conclusions were appropriately supported. However, the major area to improve readability would be to emphasize the translation of these findings into clinical applications. The manuscript can be accepted for publication after addressing minor revisions.

• The role of PD-1 for Tregs should be elaborated with relevance to graft rejection and how it is correlated with the findings of this study.

• How does PD-1 expression on CD8+CD45RClo/- and CD8+CD45RChi T cells before and after transplantation correlate with a poorer outcome? Elaborate.

• The effect of anti-CD45RC therapy on kidney transplantation outcomes should be discussed.

• A graphical abstract representing the overall findings will add readability.

• The conclusions and future direction of this study should be discussed in detail.

6. PLOS authors have the option to publish the peer review history of their article (what does this mean?). If published, this will include your full peer review and any attached files.

Reviewer #1: No

Reviewer #2: **Yes: **Vasantharaja Raguraman

---

## [Author Response · Author response to Decision Letter 0]

21 Dec 2023

The manuscript, Bézie et al., report a CD45RC T cell phenotypic signature associated with the outcome of kidney transplanted patients. This study implies the importance of biomarkers to monitor the transplant outcomes in study participant groups (STA vs. REJ). The results are very interesting, and the conclusions were appropriately supported. However, the major area to improve readability would be to emphasize the translation of these findings into clinical applications. The manuscript can be accepted for publication after addressing minor revisions. 

Answer: We thank the reviewers and editors for acknowledging the quality of our study.

• The role of PD-1 for Tregs should be elaborated with relevance to graft rejection and how it is correlated with the findings of this study. 

• How does PD-1 expression on CD8+CD45RClo/- and CD8+CD45RChi T cells before and after transplantation correlate with a poorer outcome? Elaborate. 

Answer: We added a few points to the discussion: “Interestingly, PD-1 was also reported being expressed by exhausted human CD8+HLA-DR+ Tregs (1) and dysfunctional CD4+ Tregs (2). The high frequency of HLA-DR+ cells within CD8+ and CD4+ Tregs we observed in the cohort after transplantation correlating with graft rejection supports the hypothesis of exhausted unfunctional Tregs in rejecting patients. Anti-PD-1 mAb treatment is evaluated in patients to stimulate activation of tumor reactive T cells, however it also stimulates PD1+ CD4+ and CD8+ Tregs resulting in higher Treg suppressive activity and unwanted progression of cancer (3). While anti-PD-1 mAbs might be beneficial for transplanted patients at one year post transplant to stimulate Tregs, the high frequency of activated PD-1+ (and HLA-DR+) cells in CD8+CD45RChi T cells before the transplantation correlating with poor graft outcome, argues for a later application.”

• The effect of anti-CD45RC therapy on kidney transplantation outcomes should be discussed. 

Answer: We have now added a discussion: “Targeting CD45RC with a depleting mAb showed high efficacy to establish tolerance in models of heart transplantation, GvHD, APECED and Duchenne’s dystrophy 4-7. The short-term effect was mediated by pathogenic cell depletion, while the long-term therapeutic effect in these models was mediated by Tregs of increased suppressive activity able to induce infectious tolerance. Indeed, since CD45RC is the only CD45 isoform not expressed by Tregs, it represents a promising therapeutic to target non Tregs in transplant patients.”

• A graphical abstract representing the overall findings will add readability. 

Answer: We thank the reviewer for his advice, a graphical abstract was added.

• The conclusions and future direction of this study should be discussed in detail. 

Answer: We now discuss:

“Recently developed multiproteomic analysis of pan T lymphocytes in tissue biopsies makes it possible to deeply characterize the T cell infiltrate of the graft. Our efforts will now focus on finding correlations between graft-infiltrating and blood T cells profile in order to find more precise and relevant blood diagnostic markers of the response to the allograft. In addition, a better understanding of the kinetics of response to allograft by CD8+ T cell subpopulations will allow real-time diagnosis.”

References:

1. A. Machicote, S. Belén, P. Baz, L. A. Billordo, L. Fainboim, Human CD8+HLA-DR+ Regulatory T Cells, Similarly to Classical CD4+Foxp3+ Cells, Suppress Immune Responses via PD-1/PD-L1 Axis. Front. Immunol. 9, 2788 (2018).

2. D. E. Lowther, B. A. Goods, L. E. Lucca, B. A. Lerner, K. Raddassi, D. Van Dijk, A. L. Hernandez, X. Duan, M. Gunel, V. Coric, S. Krishnaswamy, J. C. Love, D. A. Hafler, PD-1 marks dysfunctional regulatory T cells in malignant gliomas. JCI Insight 1 (2016).

3. T. Kamada, Y. Togashi, C. Tay, D. Ha, A. Sasaki, Y. Nakamura, E. Sato, S. Fukuoka, Y. Tada, A. Tanaka, H. Morikawa, A. Kawazoe, T. Kinoshita, K. Shitara, S. Sakaguchi, H. Nishikawa, PD-1 + regulatory T cells amplified by PD-1 blockade promote hyperprogression of cancer. Proc. Natl. Acad. Sci. 116, 9999–10008 (2019).

4. Boucault L, Lopez Robles MD, Thiolat A, et al. Transient antibody targeting of CD45RC inhibits the development of graft-versus-host disease. Blood Adv. 2020;4(11):2501-2515. doi:10.1182/bloodadvances.2020001688

5. Ouisse LH, Remy S, Lafoux A, et al. Immunophenotype of a Rat Model of Duchenne’s Disease and Demonstration of Improved Muscle Strength After Anti-CD45RC Antibody Treatment. Front Immunol. 2019;10:2131. doi:10.3389/fimmu.2019.02131

6. Besnard M, Sérazin C, Ossart J, et al. Anti-CD45RC antibody immunotherapy prevents and treats experimental Autoimmune PolyEndocrinopathy Candidiasis Ectodermal Dystrophy syndrome. J Clin Invest. Published online February 15, 2022. doi:10.1172/JCI156507

7. Picarda E, Bézie S, Boucault L, et al. Transient antibody targeting of CD45RC induces transplant tolerance and potent antigen-specific regulatory T cells. JCI Insight. 2017;2(3):e90088. doi:10.1172/jci.insight.90088

---

## [Decision Letter · Decision Letter 1]

9 Jan 2024

PONE-D-23-20905R1Renal graft function in transplanted patients correlates with CD45RC T cell phenotypic signaturePLOS ONE

Dear Dr. Guillonneau,

Thank you for submitting your manuscript to PLOS ONE. After careful consideration, we feel that it has merit but does not fully meet PLOS ONE’s publication criteria as it currently stands. Therefore, we invite you to submit a revised version of the manuscript that addresses the points raised during the review process.

We look forward to receiving your revised manuscript.

Kind regards,

Senthilnathan Palaniyandi, Ph.D

Academic Editor

PLOS ONE

Journal Requirements:

Additional Editor Comments:

We have now received comments from the referee of your manuscript, we invite you to submit a revised version of the manuscript. Please consider and address each of the comments raised by the reviewer.  

Reviewers' comments:

Reviewer's Responses to Questions

**Comments to the Author**

1. If the authors have adequately addressed your comments raised in a previous round of review and you feel that this manuscript is now acceptable for publication, you may indicate that here to bypass the “Comments to the Author” section, enter your conflict of interest statement in the “Confidential to Editor” section, and submit your "Accept" recommendation.

Reviewer #1: All comments have been addressed

Reviewer #2: All comments have been addressed

2. Is the manuscript technically sound, and do the data support the conclusions?

Reviewer #1: Yes

Reviewer #2: Yes

3. Has the statistical analysis been performed appropriately and rigorously? 

Reviewer #1: Yes

Reviewer #2: Yes

4. Have the authors made all data underlying the findings in their manuscript fully available?

Reviewer #1: Yes

Reviewer #2: Yes

5. Is the manuscript presented in an intelligible fashion and written in standard English?

Reviewer #1: Yes

Reviewer #2: Yes

6. Review Comments to the Author

Reviewer #1: Thanks to all authors for the intersting work. The study is intersting with pre transplantation data and a good follow up period of 4.5 years.

I have the following points:

1. In the introduction, authors stated how the current biomarkers used to predict outcome are costly and slow to implement. It would e better to reflect on how results of the current study address these issues in the discussion.

2. In the methodology, authirs stated that 50% of the cohort had AR! This is a bit high! Is there any explantaion? or it is the selection process? Better to clarify.

3. What about the pretransplant sensetization history and data of the cohort?

4. Was there any relation between the trajectory of GFR (40-60 OR >60), and proteinuria (0 or <0.5 gm) with the study results?

5. Any correlation with CNI trough levels as IS may have a differential effect in STA vs. REJ patients?

6. More elaboration about the role of PD-1 needed.

7. It would be better if there is data about any correlation with elements of humoral response or correlation with biopsy proven AMR or any MVI!?

6. Better to add the study limitation.

7. Yes, a graphical abstract will add a lot.

Reviewer #2: (No Response)

7. PLOS authors have the option to publish the peer review history of their article (what does this mean?). If published, this will include your full peer review and any attached files.

Reviewer #1: No

Reviewer #2: No

---

## [Author Response · Author response to Decision Letter 1]

16 Jan 2024

Reviewer #1: Thanks to all authors for the intersting work. The study is interesting with pre transplantation data and a good follow up period of 4.5 years.

I have the following points:

1. In the introduction, authors stated how the current biomarkers used to predict outcome are costly and slow to implement. It would be better to reflect on how results of the current study address these issues in the discussion.

Answer: We added in the discussion: “These results show that there is no dichotomous marker making it possible to predict the outcome of the transplant but a combination of markers defining the state of activation of cell subsets providing a probability on the status of the graft and of the outcome of the transplant. (…) Some markers are already clinically investigated and we propose adding at least the CD45RC marker, or even FOXP3, to add value to each of the validated markers without disrupting all the protocols.”

2. In the methodology, authors stated that 50% of the cohort had AR! This is a bit high! Is there any explantaion? or it is the selection process? Better to clarify.

Answer: Among the 300 patients included in the DIVAT biocollection, 34.7% had an AR in the first 18 months following the transplant. For this study, we selected patients such that 50% had AR and 50% did not. We clarified in the material and methods section: “This retrospective study is based on first transplanted patients selected from the Nantes DIVAT biocollection such that 50% had acute, with a rejection occurring within the first 18 months and 50% did not for 50% of the cohort.”

3. What about the pretransplant sensitization history and data of the cohort?

Answer: We indicated in Table 1 that 8.7% of selected patients were secondly transplanted.

4. Was there any relation between the trajectory of GFR (40-60 OR >60), and proteinuria (0 or <0.5 gm) with the study results?

Answer: These criteria were used to identify STA and REJ patients, so a correlation of immune markers with graft rejection assumes a correlation with these clinical markers. We defined in the material and methods section that “A stable function of the graft is defined as stable creatinine below 150 µmol/L (ideally < 100 µmol/L), zero proteinuria or less than 0.5 g/24h or g/g, an immunosuppressive treatment other than Sirolimus (Rapamune) or Everolimus (Certican), a clearance greater than 40 ml/min in MDRD and no DSA for more than one year.” 

5. Any correlation with CNI trough levels as IS may have a differential effect in STA vs. REJ patients?

Answer: The reviewer is right, all patients were treated with CNI, CsA and TAC as shown in Table 1 with doses adapted to their immune status. We added in the material and method section that “CNI doses were adapted according to the clinical criteria for graft rejection in order to minimize side effects. Therefore, patients defined as STA received lower doses of CNI than REJ patients.”

6. More elaboration about the role of PD-1 needed.

Answer: We are already discussing significantly on the role of PD-1 with new sections added following the 1st review, we believe that it would unbalance the discussion to add more elaboration on this marker only vs the other markers: “PD-1 and HLA-DR expression on CD4+ and CD8+ total T cells did not correlate with graft outcome in this cohort either, but their high expression on CD8+CD45RClo/-FOXP3+ and FOXP3- Tregs respectively after transplantation and of PD-1 on CD8+CD45RChi T cells before transplantation positively correlated with a poorer outcome. The role of PD-1 for Tregs is controversial. PD-1 has been reported being involved in CD4+Tregs generation and suppressive properties 28, to identify regulatory cells versus memory cells among CD8+ Tregs 29, and notably PD-1+CD122+CD8+ Tregs have been shown to prolong murine allograft survival 30. By contrast, other studies reported that PD-1 deficient CD4+ Tregs were more functional as inhibiting efficiently experimental autoimmune encephalomyelitis or diabetes in mice 31. (…) Interestingly, PD-1 was also reported being expressed by exhausted human CD8+HLA-DR+ Tregs 32 and dysfunctional CD4+ Tregs 34. (..) Anti-PD-1 mAb treatment is evaluated in patients to stimulate activation of tumor reactive T cells, however it also stimulates PD1+ CD4+ and CD8+ Tregs resulting in higher Treg suppressive activity and unwanted progression of cancer 35. While anti-PD-1 mAbs might be beneficial for transplanted patients at one year post transplant to stimulate Tregs, the high frequency of activated PD-1+ (and HLA-DR+) cells in CD8+CD45RChi T cells before the transplantation correlating with poor graft outcome, argues for a later application.”

7. It would be better if there is data about any correlation with elements of humoral response or correlation with biopsy proven AMR or any MVI!?

Answer: The cohort of patients was constituted in such a way as to identify markers reflecting the stable or inflammatory state of the graft such that all patients presenting humoral responses were included in the REJ group and none in the STA group. Therefore, markers correlated with stable graft function may not correlate with humoral response. We defined in the material and methods section that “Rejection was defined by humoral, cellular or borderline rejection according to Banff classification in effect at the time of diagnosis proven by biopsy. None of these signs were observed in systematic biopsy on month+12 in stable patients”. In addition, the cohort constituted does not include enough patients who developed a humoral response to study the correlation of markers with this subgroup.

6. Better to add the study limitation.

Answer: We indicated in the discussion section that “This [result] may be due to the smaller cohort of patients compared to what was used previously by other groups”, and we added “We focused our efforts on the investigation of the Teff/Treg balance and profile through broad immunophenotyping however applied to a limited number of patient samples, and were unable to increase the resolution of the prediction to the type of cellular or humoral rejection within the REJ group.”

7. Yes, a graphical abstract will add a lot.

Answer: A graphical abstract was added and submitted to PlosOne.

---

## [Decision Letter · Decision Letter 2]

21 Feb 2024

Renal graft function in transplanted patients correlates with CD45RC T cell phenotypic signature

PONE-D-23-20905R2

Dear Dr. Guillonneau,

We’re pleased to inform you that your manuscript has been judged scientifically suitable for publication and will be formally accepted for publication once it meets all outstanding technical requirements.

Kind regards,

Senthilnathan Palaniyandi, Ph.D

Academic Editor

PLOS ONE

Additional Editor Comments (optional):

Reviewers' comments:

Reviewer's Responses to Questions

**Comments to the Author**

1. If the authors have adequately addressed your comments raised in a previous round of review and you feel that this manuscript is now acceptable for publication, you may indicate that here to bypass the “Comments to the Author” section, enter your conflict of interest statement in the “Confidential to Editor” section, and submit your "Accept" recommendation.

Reviewer #1: All comments have been addressed

2. Is the manuscript technically sound, and do the data support the conclusions?

Reviewer #1: Yes

3. Has the statistical analysis been performed appropriately and rigorously? 

Reviewer #1: Yes

4. Have the authors made all data underlying the findings in their manuscript fully available?

Reviewer #1: Yes

5. Is the manuscript presented in an intelligible fashion and written in standard English?

Reviewer #1: Yes

6. Review Comments to the Author

Reviewer #1: (No Response)

7. PLOS authors have the option to publish the peer review history of their article (what does this mean?). If published, this will include your full peer review and any attached files.

Reviewer #1: No

---

## [Editor Report · Acceptance letter]

13 Mar 2024

PONE-D-23-20905R2 

PLOS ONE

Dear Dr. Guillonneau, 

I'm pleased to inform you that your manuscript has been deemed suitable for publication in PLOS ONE. Congratulations! Your manuscript is now being handed over to our production team.

Kind regards, 

on behalf of

Dr. Senthilnathan Palaniyandi 

Academic Editor

PLOS ONE